# Designing a Nurse-Led Program for Self-Management of Substance Addiction Consequences: A Modified e-Delphi Study

**DOI:** 10.3390/ijerph20032137

**Published:** 2023-01-24

**Authors:** Paulo Seabra, Inês Nunes, Rui Sequeira, Ana Sequeira, Ana Simões, Fernando Filipe, Paula Amaral, Marissa Abram, Carlos Sequeira

**Affiliations:** 1Nursing School of Lisbon, Av. Prof. Egaz Moniz, 1600-190 Lisbon, Portugal; 2Nursing Research, Innovation and Development Centre of Lisbon (CIDNUR), Av. Prof. Egaz Moniz, 1600-190 Lisbon, Portugal; 3Center for Health Technology and Services Research (CINTESIS), Faculty of Medicine, University of Porto, R. Dr. Plácido da Costa, 4200-450 Porto, Portugal; 4Specialized Technical Treatment Unit of Barreiro—Integrated Responses Center, Avenida do Bocage n°34, 2830-002 Barreiro, Portugal; 5College of Nursing and Public Health, Adelphi University, One South Avenue, P.O. Box 701, Garden City, NY 11530-0701, USA; 6Nursing School of Oporto, Rua Dr. António Bernardino de Almeida, 4200-072 Porto, Portugal

**Keywords:** substance-related disorders, self-management, nursing, program evaluation, Delphi study

## Abstract

Therapeutic interventions for people with problematic use of psychoactive substances can help tackle specific needs related to substance addiction consequences. This modified e-Delphi study aimed to establish consensus on a training program for self-management of substance addiction consequences. The study was conducted between February and April 2022, with an experts’ sample of 28 participants in the first round and 24 in the second. A priori consensus criteria were defined for each round. The results revealed a very strong consensus was achieved on the structure of the program and on clinical areas, such as the problematic use of substances, general health knowledge, health-seeking behavior and adherence, self-knowledge and well-being, social role and personal dignity, and family process. Additionally, over 80% participant consensus was achieved on an extensive number of interventions categorized as psychoeducational, psychotherapeutic, socio therapeutic, brief interventions, social skills training, problem solving techniques, relaxation techniques, and counseling. These findings may be able to fulfill a gap concerning structured treatment approaches for people with problematic use of psychoactive substances. Supporting self-management of the consequences of substance addiction and its application can change nurses’ interventions.

## 1. Introduction

Problematic substance use (PSU) and addiction are pervasive global health issues. In 2021, the United Nations Office on Drugs and Crime (UNODC) reported that 275 million people used drugs, while 36 million people had an addiction to a psychoactive substance worldwide [1]. PSU occurs when an individual begins to experience negative consequences related to substance use, which can lead to addiction. Addiction (also known as substance use disorders) is defined as a “chronic, relapsing disorder that changes brain function and structure” [2]. It is characterized by compulsive drug seeking and use despite adverse consequences [2], impairing an individual’s ability to effectively self-manage [3]. Substance addiction consequences (SAC) are a common reason for an individual to seek care and enter treatment. Self-management is the ability to adapt and self-manage in the face of social, physical, and emotional challenges [4]. This is important as addiction is a treatable disorder and recovery can be supported by evidenced-based interventions. Self-management can support an individual’s recovery by helping them to manage the impact of SAC.

Individuals with PSU and addiction are entitled to the highest standard of health care. Treatment programs should include comprehensive assessment, psychosocial interventions, and pharmacological interventions to manage mental and physical health comorbidities within the context of social care and support. Furthermore, active patient participation is essential in all treatment decisions [1]. Nurses, all over the world, are at the forefront of responding to the needs of people with PSU and addiction. To provide evidenced-based care, nurses must contribute to the identification of intervention programs that respond to people’s needs, based on the best available scientific evidence [5], but when it comes to “nursing intervention programs” for individuals with substance use, the evidence is still scarce [6].

There is increasing evidence that structured therapeutic interventions for people with SAC can be of benefit and help tackle specific symptoms and problems [7,8]. Different interventions have been used by healthcare professionals, in modalities such as groups or individuals [9]. The most frequently used interventions are motivational strategies [10], cognitive behavior therapy [8], and psychoeducation for relapse prevention [7]. However no interventions or nurse-led programs have been found to address the self-management of addictions’ consequences [11].

Understanding the importance of self-management in addiction treatment and recovery supported by a robust theoretical basis, we created an evidence-based nurse-led training program. The program focused on the self-management of substance addiction consequences for people integrated on medication-based programs using the modified e-Delphi technique. This program was based on different theoretical foundations.

### 1.1. The Tidal Model

From a theoretical point of view, we integrated the principles of the Tidal Model, which considers that human nature can be expressed through physical, emotional, intellectual, social, and spiritual planes. The Tidal Model is a patient-centered model that is centered on recovery focused care. A key assumption is “tailoring care to fit the specific needs of the person by understanding the person’s story and unique lived experience told through a holistic assessment” [12] [p.173]. The individual maintains autonomy while the nurse “negotiates” support and promotes personal security. Although a collaborative relationship, the individual assumes an active role in their own treatment, which is focused on facilitating self-management. The recovery approach of this model aims to help people recover their life story, recognizing that they are more knowledgeable about themselves and their needs [13]. The Tidal Model helps people strengthen their awareness about the small changes that will have a significant effect on their lives, in which empowerment is the central focus. All these principles can be linked to the recommendation that addiction treatment and recovery programs must have goals based on people’s needs. Furthermore, these goals should be clear, realistic, measurable, consider risk and protective factors, and be formulated according to the changes expected for users [14].

### 1.2. Substance Addiction Consequences (SAC)

The severity of substance addiction consequences (SAC) is often the reason for seeking help and joining medication-based programs (methadone, buprenorphine, and alcohol aversion therapy). It can be defined as the severity of change in health status and social functions due to substance dependence [15]. SAC is a concept related to the impact felt by people with problematic substance use and addiction problems. SAC can impact the individual on different personal and social dimensions (psychological and family, physical and cognitive, economic and labor, and self-care) and can be helpful to develop nursing diagnoses. SAC is a person-centered assessment tool that can measure anxiety, sadness, difficulty maintaining employment, problems with self-supporting, and problems in family relationships. These problems are related to physical or psychological comorbidities [3], the frequent trauma experienced on the past or as consequence of substance use [16], and the impact on family functioning and emotional status [17]. These experiences can disrupt the individual’s ability to effectively self-manage and recover from problematic substance use and addiction.

### 1.3. Chronic Illness Self-Management

Problematic substance use can lead to addiction, a chronic illness caused by changes in brain functioning that affect areas critical for judgment and self-regulation [18]. Chronic Illness Self-Management is a strategy for individuals to manage their chronic conditions. It teaches them to actively identify challenges and solve problems related to their illness [4]. Chronic Illness Self-Management can be characterized into three dimensions. The first concept is related to the person, in which it is considered that patients should actively participate, take responsibility for the care process, and have a positive way of facing adversity. The second focuses on the person’s relationship with the care environment, in which it is considered that the person should be informed about the condition, illness, and treatment, and the self-management process must be individualized. It requires openness to ensure a reciprocal partnership with professionals and requires openness to social support. Finally, the third component recognizes that managing chronic illness is a lifelong task that requires personal skills (decision-making and problem-solving skills) to maintain health. Living with a chronic illness often requires programs to train and develop a better capacity for self-management of SAC [11,19].

Together, the Tidal Model and Chronic Illness Self-Management can empower the individual to change and to improve the management of the chronic problematic relationship with substances. This can be utilized with the trans-theoretical model, which can help the nurse understand how the individual is progressing through the health behavior change process [20]. The process of change is facilitated by different techniques, methods, and interventions usually associated with different theoretical orientations. From an interventional point of view, it requires different psychotherapeutic approaches [21], such as motivational [20] and brief interventions [22]. These psychotherapeutic interventions structured on the Psychotherapeutic Intervention Model in Nursing [23] demonstrate the need to develop a structure (number of sections, time of the sections, and time between them), process (assessment framework, nursing diagnoses and interventions), and outcomes (nursing outcomes) for nursing psychotherapeutic person-centered care. These theories were integrated to support the design of the program.

The aim of this study is to identify and establish consensus on a training program for self-management of substance addiction consequences (in terms of content and form).

## 2. Materials and Methods

### 2.1. Design

The initial draft of the program was based on a scoping review that examined evidence of structured therapeutic interventions for people with substance addiction consequences [24] and a qualitative study exploring both the experience of caring for an individual using substances or having addiction [19]. At this stage of development of the intervention, the draft specifically focused on modeling the processes and outcomes [25]. To establish consensus, we developed a modified e-Delphi study [26], an online survey that contained closed ended items and open comment boxes in which we addressed the following research questions:-How do participants rate the importance of each part of the program?-Which parts of the program are considered essential by participants?-What suggestions do participants add to the structure, areas of focus and interventions?

The main assumption of the Delphi technique is that expert group consensus is more valid than individual analysis. It is useful for achieving agreement among experts or professionals in something where no previously agreement existed [27]. Because we used an online survey, this study is considered a modified e-Delphi study [28]. Guidelines for Conducting and Reporting of Delphi Studies (CREDES) were followed [28]. Before we sent our online survey, it was piloted on six addiction nurses who detected some minor inaccuracies in the platform, but not in the content. Participants were invited via an initial e-mail to participate in the study. After accepting, a second email was sent with a flowchart of the process and a description of the program including detailed and concrete information such as objectives, admission criteria, context of the intervention, sequence of sessions, diagnoses and nursing interventions, and a link with the final survey elaborated in Google Docs, asking for a response in 14 days. Experts were informed that the number of rounds necessary would be determined by the level of consensus [27].

### 2.2. Participants

We utilized intentional non-probabilistic sampling. The selection criteria included the following: (1) Nurses AND (2a) Professional experience of at least 2 years with patients with problematic use of psychoactive substances or addiction in outpatient care, in the last 10 years OR (2b) Professional experience of at least 5 years, in the last 10, in inpatient psychiatric wards, with patients with substance use comorbidity OR (2c) Relevant curricular experience in the area of mental health or community health and experience in the construction of intervention programs with people with health needs in these two areas (provable by scientific publications, theses, or dissertations). Expertise was defined by professional and clinical experience [28], the length of time working with patients with problematic use of substances and addiction, and the experience in construction of interventions programs.

There is no agreement in the literature about the ideal sample size for Delphi studies, although usually 20 participants are considered sufficient. It is dependent on the purpose of the project, the time frame for data collection, and the number of rounds. Twenty nurses was considered the minimum for each round to ensure diversity of opinions [28]. Invitations were sent to 35 nurses, with 28 responses (80%) in the first round and 24 (69%) in the second round. To maintain the heterogeneity and rigor of expert opinion a value of 70% was the minimum acceptable of responses rate (rate on the second round was slightly under this value) [29].

### 2.3. Data Collection

Data were collected between 14 February and 3 April 2022. The first round was necessary in order to complete consensus on the structure and content of the program and a second round was added to obtain consensus on the expert feedback that was given from the open-comment boxes in round 1.

Data were extracted from the Google Docs platform to an Excel file protected with a password on the computers of two researchers. Individual data were collected (age, profession, academic degree, previous experience working on addictions, and previous experience on design of interventions on mental health and communitarian health). Each participant was invited to rate how much he/she agreed on the structure of the proposed program (number, interlude and time of sessions, context, and generic articulation procedures between sessions), and areas of intervention/nursing diagnoses/interventions. A five-point Likert scale (1—not suitable; 2—poorly suitable; 3—moderately suitable; 4—very suitable; 5—totally suitable) was used. Omitting a labeled neutral mid-point forces participants to deliberate and form an opinion [30]. Through open comment boxes, participants were invited to identify and add any important missing clinical indicators or to explain the reasons for the ratings of their judgments [31].

### 2.4. Data Analysis

There is no full agreement on the consensus rate, but 70% is considered adequate [28,31]. For consensus, a median (M) value of participants’ Likert scale ≥3.5 (with standard deviation (SD) ≤ 1 and interquartile range (IQR) ≤ 1) and percentage of agreement on 4–5 ≥ 80% was needed in our study. Constructs were not included on the next round if the consensus fell between the lowest three scale points (not suitable to moderately adequate). For the comment box data, the content was analyzed by the research team. Items with participant suggestions for improvement were added after analyses of the responses and were aggregated by the meaning [27]. After the first round, a new survey, with concrete items, was sent to participants, informing participants about the content in which the consensus was obtained.

Quantitative data were analyzed using the IBM Statistical Package for Social Sciences (SPSS) v27.0. Frequencies, mean (M), median (MD), standard deviation (S), and interquartile range (IQR) were calculated to assess consensus. Additional recommendations by experts were evaluated by the research team to confirm if they were not already covered by the survey, within the scope of the study, and articulated clearly. Where they were not, the research group reviewed and adjusted the content [30]. These additional constructs were then added to second round, after being reviewed and piloted with research team members not included on data analyses, to prevent bias [28]. The final draft was intensively discussed and reviewed by the research team (academic and clinical nurses) to approve its final version.

### 2.5. Ethical Considerations

This study had the previous approval of the Health Ethics Committee of the Regional Health Administration of Lisboa and Vale do Tejo (7211/CES/2020). All participants were sent an initial invitation in which they could access the link to the server, read the informed consent, and they had the possibility to refuse or accept their participation. Until the last question and before submitting, participants could just close the survey and not submit it. The platform and data files were protected with a password, known only by two researchers.

## 3. Results

Participants (n = 28) were mostly women 21 (75%), with a mean age of 42.1 years (SD = 7.84) (range 26–54), with a similar representative academic degree on master or doctoral studies [12] and bachelor’s degree in nursing [4]. Regarding nursing credentials, 2 were Registered Nurses, 23 Specialists in Mental Health and Psychiatric Nursing, and 3 Specialists in Public Health. Half of the participants were clinicians and the other half were academics or researchers, and 18 had previous experience in designing intervention programs. Participants had a mean of 19.0 years of professional experience in clinical nursing (SD = 8.15) and a median of 5.50 (IQR = 15) years of clinical experience with patients with problematic use of substances. Participants in this study were from Portugal.

### 3.1. Round 1

The consensus was achieved in all parts of the program content, previous procedures, therapeutic sessions, and final procedures. All median values ≥4, IQR ≤1 and 80% of agreement in 4 or 5 on rate scale (very or totally suitable) (Table 1).

Through open boxes for each content evaluation, participants could add comments or suggestions, identifying missing important clinical indicators or explaining the reasons for the chosen ratings. Most suggestions concerned the need to formalize (possible) nursing diagnostics, using the International Classification for Nursing Practice (ICNP) [32]. In fact, the initial proposal only focused on needs and areas, without a rigorous formalization on the nursing classified language systems. Other suggestions were to redefine the objectives; reduce the length of time for each session, based on the brief intervention’s theoretical orientation; clarification if capacity to read was an inclusion criterion for participants; clarification if cognitive impairment was a strict exclusive criterion; definition of help relationship as the main clinical orientation; and clarification of procedures in the previous evaluation session, initial session, following sessions, and final session.

### 3.2. Round 2

The second round happened between the end of March and April 2022. Based on the participants’ responses, small but important changes had to be made in almost all areas of the program. Therefore, we decided to resend the revised detailed description program, as well as its schematization, and ask for it to be reassessed entirely, aiming to also evaluate the answers’ stability between rounds—knowing that there should be a less than 15% change in the responses [30]. Two more closed ended items were added to the second round. Because of the doubts raised by the participants, it was necessary to introduce a question about the educational level and time of experience required for nurses who will implement the program. In addition, an additional question was added: “Do you consider appropriate to finalize the process of defining terminological definitions of diagnoses and interventions according to the nursing classified languages, after experts’ consensus and during clinical validation?”.

Again, consensus was achieved in all parts of the program content (except on the time between sections, 79.2%, in which we made no changes), previous procedures, therapeutic sessions, and final procedures (with no consensus in the proposed postponed final terminological definition of diagnoses and interventions according to the systems of classified language, 79.2%). All of them had a median ≥4 and IQR = 1 (Table 2).

Between rounds, the difference in the mean of all items was 0.08 (first round 4.28 and second round 4.36), and the difference of median values was 0.07 (first round 4.24 and second round 4.35). In agreement, the percentage the difference between the two rounds was 0.08 (first round 90.04 and second round 89.96). These three criteria had a strong and stable consensus, with the average change between two rounds being less than 15%.

## 4. Discussion

The consensus reached after the expert opinion poll led to the conclusion that a nurse-led program for self-management of substance addiction consequences is pertinent and necessary to support an individual’s needs. Our results show that a nurse-led model of addiction treatment must be flexible enough to incorporate local needs and be transferable enough to be applied in the face of various normative guidelines and practice settings. It must be capable of being implemented with fidelity and sustainably, and allow for an evaluation of the process and impact [14]. In the sparse literature on the subject, the conclusions support our results. The nurse-led intervention must be comprehensive and focus on an individual’s identified needs. It should have an adapted duration, be measurable and implementable, consider environmental factors, have a biopsychosocial approach, and may utilize brief interventions to achieve observable behavioral changes [6]. Unfortunately, the few available treatment programs in the literature did not seem adequate to provide comprehensive specialized care for people integrated in medication programs [6,8,33]. In addition, the focus on the self-management of consequences seemed poorly explored [34]. In our study, there was a high consensus about the role self-management in the recovery process.

Participants rated the importance of each part of the proposed program and identified parts that were considered essential: a structured but flexible program. For example, a structured approach should consider selection criteria, number, time of sessions, and established procedures for all sessions, whereas flexible attention is needed to support substance addiction consequences experienced by patients. These procedures and clinical orientation follow the recommendations for the development of a complex intervention [25]. The suggestions added on the structure, clinical areas, and interventions, based on participants’ open answers, were mostly based on the need to formalize nursing diagnostics using International Classification for Nursing Practice (ICNP) classified language [32]. The rate on the second round was 79.2%, because some participants argued that this process should be based on Nursing Ontology [35], another classified language adopted from ICNP in Portugal, which is in an early implementation stage. Another important component, the time interval between sessions, had a difference of opinion, as some of the experts preferred less time between them, based on a psychotherapeutic point of view. However, we decided to keep it flexible, as this was supported by other studies [36], with a maximum interlude of 3 weeks. Less of this flexible interlude can be a barrier to the development of the program.

The theoretical framework guiding assessment and intervention were based on a holistic view of the person and on their ability to be helped to reach their full potential using their own personal resources [11,13]. It also recognized that intervention must be based on assessing and managing the consequences of substance dependence [3]. To do this effectively and help individuals with PSU and addiction, it is important to enhance motivation for change [20]. Therefore, the principles of motivational interviewing are an essential therapeutic strategy [20].

Intervention models for addiction treatment must have an integrative approach; that is, allowing the use of psychotherapeutic techniques with different approaches and based on an effective relationship and interpersonal communication between the nurse and the person being cared for [23]. Intervention programs should raise awareness about each person’s resources, guide them towards healthy alternatives, and support them in caring for their own health in a health-promoting environment. Activities must be time-planed and adapted to the characteristics of the target population. The intervention model should, whenever possible, have a timetable that promotes adherence and have particular attention to gender issues [14], the trauma lived experience [16], and family centeredness and involvement [17], because all of these factors have an impact on substance addiction, with consequences on areas such as emotional status, social role, and personal dignity. These aspects were considered when defining the structure and therapeutic strategies mobilized in this program.

Our proposal is adapted, in structural terms, from the Psychotherapeutic Intervention Model in Nursing [23], but also includes other contributions in its content that are highlighted throughout. The structure and monitoring of processes follows other international recommendations for programs in the area of addictive treatment [14]. The intervention areas (focus/diagnoses) are based on previously carried out studies: characterization of the population [3], opinion of professional experts and individuals with lived experience of addiction [19], and a scoping review [24]. The selected areas of intervention are related to problematic use of substances; general health knowledge; health-seeking behavior and adherence; self-knowledge and well-being; social role and personal dignity; and finally, the family process. We achieved consensus on an extensive number of interventions—psychoeducational, psychotherapeutic, socio therapeutic, brief interventions, social skills training, problem solving techniques, relaxation techniques and counseling—that are also used in other interventions, which have an impact proven by research [7,8], although in this study there was the specific purpose of supporting self-management.

The program focuses on the needs expressed by individuals with PSU, addiction, and nurses in outpatient medication-based programs. Operationalization appears in Appendix A “Programa para a Autogestão da Dependência de Substâncias” (ADS Program). The goals of the program are to train and empower people to self-manage the substance addiction consequences, and improve adherence to their therapeutic project. It was considered that this program can be applied in other countries where nurses have the same level of skills and competence (working as mental health or community specialist nurses or a registered nurse with experience in addiction and under the supervision of a mental health specialist nurse for psychotherapeutic interventions) and where the organization of health services is similar (outpatient-center-based and with legislation that allows for a comprehensive assessment and intervention even when patients are not abstinent). However, it should be noted nurse-led intervention proposals can be influenced by issues such as environmental, structural, and organizational factors. In addition, personal, professional, and multidisciplinary issues such as a lack of education in relation to recovery, harm reduction, and trauma inform mental health care, group culture, education and training, and public attitudes towards addiction and addiction treatment can impact the treatment of PSU and addiction.

### Limitations

A possible limitation of this study was the need to clarify the concept of substance addiction consequences, which was new to almost all of the participants, and to develop structured interventions addressing this problem. Some of the suggestions were not integrated because they were beyond the concept that this program intended to address. Because this study was conducted in Portugal, it may not be generalizable to different backgrounds and cultures. Additionally, because we utilized an electronic survey, participants had to submit their full answers at the same time, without the possibility to respond in parts.

## 5. Conclusions

This study aims to fill a gap in scientific knowledge concerning structured approaches for people with PSU and addiction. This nurse-led intervention program aims to contribute to the improvement of the quality of life of people who use substances and are integrated into medication programs. The study can be considered to be a contribution due to the number, heterogeneity, and plurality of participants in each round. Six clinical areas and eighteen nursing diagnoses were considered in the e-Delphi: (1) problematic use of substances (diagnoses: substance abuse, compromised withdrawal behavior, and acceptance of compromised health status), (2) general health knowledge (diagnoses: insufficient health knowledge), (3) health-seeking behavior and adherence (impaired disease self-management and compromised attitude towards the therapeutic regimen); (4) self-knowledge and well-being (compromised self-image, stigma, anxiety, sadness, and loneliness), (5) social role and personal dignity (compromised autonomy, ability to manage finances compromised, committed socialization, social isolation, deficit in self-care (personal hygiene), and difficulty in having a functional home and (6) family process (compromised family process).

This proposal, being flexible and anticipating different characteristics of contexts and patients with PSU and addiction, needs to be tested in a real environment, complying with all ethical procedures. It can also be of clinical utility, after outcome analyses, to explore the potential to be adopted in different treatment contexts, not only in an outpatient setting.

## Figures and Tables

**Table 1 ijerph-20-02137-t001:** Rate and consensus in first round.

Item	Mean	Standard Deviation	Median	IQR	Agreement [4–5] ≥80%
1—Expected positive results with the program	4.68	0.548	5	1	96.4%
2—Objectives of the program	4.32	0.772	4	1	89.3%
3—Inclusion Criteria	4.41	0.636	4	1	92.5%
4—Context of the program	4.54	0.576	5	1	96.4%
5—Initial evaluation of the patient	4.44	0.641	5	1	92.6%
6—First session (exploration)	4.43	0.634	4.5	1	92.9%
7—Previous procedures to all intermediate sessions	4.46	0.576	4.5	1	96.4%
8—Flexible organization between all intermediate sessions	4.39	0.737	5	1	85.7%
9—End procedures between all intermediate sessions	4.54	0.508	5	1	100%
10—Procedures for final session (empowerment)	4.33	0.620	4	1	92.6%
11—Number of sessions (8)	4.15	0.662	4	1	85.2%
12—Duration of sessions (20–50 min)	4.36	0.621	4	1	92.9%
13—Break between sessions (1–3 weeks)	3.96	0.599	4	0	80.8%
14—Methods, pedagogical techniques, and didactic resources	4.37	0.492	4	1	100%
15—Adequacy to the outpatient setting	4.23	0.514	4	1	96.1%
16—Suitability to patients needs	4.26	0.656	4	1	88.9%
17—Adequacy of the evaluation/monitoring strategy (documentary process)	4.20	0.577	4	1	92%
18—Adequacy of diagnoses and interventions “Referring to problematic substance use”	4.07	0.781	4	1	81.5%
19—Adequacy of diagnoses and interventions “Related to health knowledge in general”	4.21	0.686	4	1	85.7%
20—Adequacy of diagnoses and interventions “Related to health-seeking behavior and adherence”	4.18	0.670	4	1	85.7%
21—Adequacy of diagnoses and interventions “Related to self-knowledge and well-being”	4.04	0.744	4	1	82.1%
22—Adequacy of diagnoses and interventions “Related to social role and personal dignity”	4.04	0.744	4	1	84.6%
23—Adequacy of diagnoses and Interventions “Related to family support”	4.11	0.629	4	1	85.7%
24—Adequacy of the areas of attention (focus/diagnoses) in the intermediate sessions	4.11	0.685	4	1	82.2%
25—Adequacy of therapeutic interventions (general, psychotherapeutic, psychoeducational, psychosocial, and socio therapeutic) in the intermediate sessions.	4.18	0.548	4	1	92.9%

**Table 2 ijerph-20-02137-t002:** Rate and consensus in the second round.

Item	Mean	Standard Deviation	Median	IQR	Agreement [4–5] ≥80%
1—Expected positive results with the program	4.67	0.565	5	1	95.8%
2—Objectives of the program	4.38	0.647	4	1	91.6%
3—Inclusion criteria	4.46	0.588	4.5	1	95.8%
4—Context of the program	4.42	0.584	4	1	95.8%
5—Initial evaluation of the patient	4.50	0.590	5	1	95.9%
6—First session (exploration)	4.50	0.590	5	1	95.9%
7—Previous procedures to all intermediate sessions	4.50	0.590	5	1	95.9%
8—Flexible organization between all intermediate sessions	4.46	0.588	4.5	1	95.8%
9—End procedures between all intermediate sessions	4.54	0.588	5	1	95.8%
10—Procedures for final session (empowerment)	4.38	0.647	4	1	91.6%
11—Number of sessions (8)	4.21	0.658	4	1	87.5%
12—Duration of sessions (20–50 min)	4.29	0.859	4.5	1	83.3%
13—Break between sessions (1–3 weeks)	4.17	0.761	4	1	79.2%
14—Methods, pedagogical techniques, and didactic resources	4.46	0.588	4.5	1	95.8%
15—Adequacy to the outpatient setting	4.38	0.647	4	1	91.6%
16—Suitability to patients needs	4.38	0.647	4	1	91.6%
17—Adequacy of the evaluation/monitoring strategy (documentary process)	4.17	0.702	4	1	83.3%
18—Adequacy of diagnoses and interventions “Referring to problematic substance use”	4.50	0.549	5	1	95.9%
19—Adequacy of diagnoses and interventions “Related to health knowledge in general”	4.33	0.868	5	1	83.4%
20—Adequacy of diagnoses and interventions “Related to health-seeking behavior and adherence”	4.38	0.711	4.5	1	87.5%
21—Adequacy of diagnoses and interventions “Related to self-knowledge and well-being”	4.21	0.833	4	1	83.4%
22—Adequacy of diagnoses and interventions “Related to social role and personal dignity”	4.29	0.751	4	1	83.3%
23—Adequacy of diagnoses and interventions “Related to family support”	4.29	0.690	4	1	87.5%
24—Adequacy of the areas of attention (focus/diagnoses) in the intermediate sessions	4.29	0.690	4	1	87.5%
25—Adequacy of therapeutic interventions (general, psychotherapeutic, psychoeducational, psychosocial, and socio therapeutic) in the intermediate sessions.	4.38	0.576	4	1	95.9%
26—Adequacy of nurse’s experience and education requirements	4.13	0.947	4	1	83.3%
27—Possibility to postpone the final process of defining the terminological definition of diagnoses and interventions according to the systems of classified language.	4.17	0.868	4	1	79.2%

## Data Availability

The data presented in this study are available upon request from the corresponding author. The data are not publicly available due to ethical considerations.

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
