# Peer review of "Designing a Nurse-Led Program for Self-Management of Substance Addiction Consequences: A Modified e-Delphi Study"

_ijerph, 2023, doi:10.3390/ijerph20032137_

Round 1

Reviewer 1 Report

IAuthors mentioned that "The aim of this study is to identify and establish nursing consensus on a training program for self-management of substance addiction consequences [in terms of content and form].". But why are they needed to identify and establish nursing consensus on a training program for self-management of substance addiction consequences? Without nursing consensus, dose it cause a problem that therapists would use a training program for self-management of substance addiction consequences?

Reviewer 2 Report

The manuscript “Designing a nurse-led training program for self-management of substance addiction consequences: An e-Delphi” describes the development process of the e-Delphi to be develope a nurse-led training program for self-management of substance addiction consequences. While the study has merit and represents a major step towards the need-based support of addiction patients, there are several limitations and shortcomings which need to be addressed before acceptance:

1) It remains unclear to me from the 3 questions (lines 148-150) and subsequent tables of results how detailed the given response options were for study participants. Were response options given by the Reasearcher team that were then scored? As an example, were multiple answer options given in terms of time and it ended up that 80.8% of participants chose 1-3 weeks between sessions? It would certainly be informative to include a document with the items in the appendix to avoid ambiguity.

2) The tables should be adjusted again to optimize the presentation. Especially the formatting in relation to the headings is unfortunate. If you drag the first column "Item" a bit smaller, the short headers "Median" and "Mean" will not be split on different rows. In addition, the heading of the last column should also be split between the rows in a more sensible way. Your valuable work deserves a pretty presentation :-)

3) As a limitation, they should also include the technical hurdles of an e Delphi.

4) In line 350 you have used an incorrect bracket, which I would ask you to adjust.

5) In lines 108, 141, 156, 295, 312 and 331, please remove the superfluous spaces.

6) The document in the appendix seems cluttered. A clearer presentation is necessary.

I look forward to your reply and wish you success for the resubmission

Reviewer 3 Report

This study aimed to establish consensus on a training program for self-management of substance addiction consequences. A modified e-Delphi study was conducted. The results revealed a very strong consensus was achieved on the structure of the program and on clinical areas, such as problematic use of substances, general health knowledge, health-seeking behavior and adherence, self-knowledge and well-being, social role and personal dignity, and family process. Over 80% participants’ consensus was achieved on an extensive number of interventions. These findings may be able to fulfill a gap concerning structured approaches, with people with problematic use of psychoactive substances. It was achieved a high consensus regarding its flexible structure and clinical areas, considered as important to support self-management of the consequences of substance addiction.

A possible limitation of this study was the need to clarify the concept of substance addiction consequences, which was new to almost all of participants, and develop structured interventions addressing this problem. This study was conducted in Portugal it may not be generalizable to different backgrounds and cultures. The Authors have correctly described these issues.

The methods are clear and well detailed. The conclusions are adequately supported by the results and interpretation of the data. The references are appropriate and current. Tables 1 and 2 are clear. Appendix A is interesting.

The introduction is a bit short, but enough.

English can be improved.

Round 2

Reviewer 1 Report

Authors replied that "It was because the program was designed to be implemented by nurses. We withdraw that statement." Deleting that statement could not solve study's limitation. This study does not contain any special findings. There is a problem that it is not able to give a new insight away from the existing framework.